# Observability of Latent States in Generative AI Models

## Abstract

We tackle the question of whether Large Language Models (LLMs), viewed as dynamical systems with state evolving in the embedding space of symbolic tokens, are observable. That is, whether there exist distinct state trajectories that yield the same sequence of generated output tokens, or sequences that belong to the same Nerode equivalence class ('meaning'). If an LLM is not observable, the state trajectory cannot be determined from input-output observations and can therefore evolve unbeknownst to the user while being potentially accessible to the model provider. We show that current LLMs implemented by autoregressive Transformers are observable: The set of state trajectories that produce the same tokenized output is a singleton, so there are no indistinguishable state trajectories. But if there are 'system prompts' not visible to the user, then the set of indistinguishable trajectories becomes non-trivial, meaning that there can be multiple state trajectories that yield the same tokenized output. We prove these claims analytically, and show examples of modifications to standard LLMs that engender unobservable behavior. Our analysis sheds light on possible designs that would enable a model to perform non-trivial computation that is not visible to the user, as well as on controls that the provider of services using the model could take to prevent unintended behavior. Finally, to counter the trend of anthropomorphizing LLM behavior, we cast the definition of 'feeling' from cognitive psychology in terms of measurable quantities in an LLM which, unlike humans, is directly measurable. We conclude that, in LLMs, unobservable state trajectories satisfy the definition of 'feelings' provided by the American Psychological Association, suitably modified to remove self-reference.

## 1 Introduction

Generative models of tokenized sequential data, including large language models (LLMs), can be interpreted as discrete dynamical systems with a latent state (sometimes referred to as 'mental state') that is updated at each step by processing an input token to produce a new output token. For example, autoregressive (AR) Transformer-based LLMs co-opt a sliding 'context' window of data as the state, and simply feed back the latest output token to the input, discarding the oldest token. State space models (SSMs) maintain a separate and explicit latent state, distinct from their input and output Zancato et al. (2024). Either way, this paper is concerned with the study of the latent state of LLMs understood as dynamical systems.

The three pillars in the analysis of dynamical systems are controllability, stability, and observability. Controllability is concerned with the existence of functions of the output that, if fed back as input in closed loop, result in a desired state sequence, or 'trajectory'. Controllability of LLMs has been studied by Soatto et al. (2022), and is relevant to the safe operation of AI bots. Stability is concerned with state trajectories remaining bounded and non-degenerate. Stability of LLMs interacting with humans in closed loop has been studied by Marchi et al. (2024). Observability is concerned with the existence of distinct state trajectories that produce the same output trajectory; consequently, there are different trajectories that are indistinguishable when one can only measure inputs and outputs. For LLMs, lack of observability would imply that there exist mental state trajectories that evolve unbeknownst to the user. Such trajectories can be triggered by input observations (prompts) or by the model's own output in closed loop. The existence of unobservable dynamics could be concerning if exploited maliciously by users intent to turn LLMs into Trojan Horses (Hubinger et al., 2024).

Further, a non-observable model could store information in a manner that is not accessible to the user, yet could be accessed by the operator.

To our best knowledge, this paper is the first to tackle the analysis of observability of LLMs. While analysis does not produce methods to mitigate undesirable behavior, it can *inform* the limits of such methods, and therefore their design.

In this paper, we show that the 'mental state' of AR LLMs is trivial (a singleton) and uniquely determined by their measurable output. That means that there are no distinct trajectories that produce the same output, or equivalently that, from the output, one can uniquely recover the latent[1] state. However, when the model is supplied with 'system prompts' not visible to the user, the state is generally unobservable. More specifically, we (i) formalize the problem of observability for LLMs, (ii) show that current autoregressive Transformer-class LLMs trained with textual tokens are observable, and (iii) show that simple modifications of the architecture can render the LLM unobservable.

ON THE COGNITIVE PSYCHOLOGY OF LLMS

Our analysis of the observability of LLMs was prompted by a simple question: Can LLMs have 'feelings'? LLMs are increasingly described in anthropomorphic terms, not just in the press or popular discourse, but also in scientific and policy writings. Methods from cognitive psychology are increasingly used to analyze LLMs as if they were opaque, observing their expressed output in response to designed inputs. The use of terms from folk psychology helps describe their behavior in ways that are evocative and familiar to most readers.

However, the lack of precise definitions of these terms in relation to LLMs often results in polarized positions with no verifiable or falsifiable correlates. While the human brain is not directly accessible for observation, LLMs are engineered artifacts every aspect of which is measurable, at least to the designer and trainer of the model. From the data to the loss function to the optimization procedure, to the activations and the weights in the trained model, to the system prompts. Every trained weight can be measured, and every activation component in response to a prompt can be observed and modified if so desired. This is quite unlike the human brain.

Therefore, when employing terms from cognitive psychology to describe LLMs, such terms should be uniquely defined and correspond to a constitutive component of the trained LLM which can be directly measured. In this paper, we focus on the notion of 'feelings' as defined by the American Psychological Association (APA Definition). Since the APA Definition is written in terms of other concepts, those must in turn be uniquely defined, going down the tree until the concept of 'feeling' is unambiguously mapped to specific components of an LLM, without creating 'loops.' We explore the APA Definition and modify it to avoid circularity. A consequence of the analysis in this paper is that, according to the APA Definition, 'feelings' in LLMs are unobservable trajectories. One could therefore tackle the question of whether a particular LLM can have feelings by analyzing its observability properties. The answer hinges on the definition, so this paper devotes a considerable amount of discussion to it. The reader interested only in the technical aspect of LLM observability, or on the security implications of lack of observability, can skip these sections. The reader interested in the analysis of 'meanings', which is a prerequisite for the definition of 'feelings', can refer to Liu et al. (2024); Soatto et al. (2022).

### 1.1 RELATED PRIOR WORK

When viewing LLMs as dynamical models, the three key properties to be analyzed are their *stability*, *controllability*, and *observability*. The analysis of LLMs viewed as dynamical models has been championed by (Soatto et al., 2022) and first focused on their *controllability* (Bhargava et al., 2023; Soatto et al., 2022). Their *stability* when operating in closed-loop has been studied by Gerstgrasser et al. (2024); Marchi et al. (2024). To the best of our knowledge, our work is the first to analyze their *observability*. Our scope falls within the broad and fast-growing area of *interpretability* of LLMs, only part of which relevant to our goal; we limit our survey to prior work using tools from systems and control theory. As our work touches upon controversial topics, such as 'meanings' and 'feelings';

---

[1]One should not confuse 'latent' with 'unobservable' in reference to a state trajectory: Latent just means that a state is not directly *observed*. However, a latent state may or may not *observable*.

we restrict our attention to minimalistic definitions pertaining to LLMs, with no implications for psychology and cognitive science. Finally, since the ramifications of our analysis touch upon the issue of safety and security of LLMs, we briefly survey relevant literature as it relates to our contribution.

**Capabilities of LLMs.** As LLMs blaze through tests meant to measure human cognitive abilities, from the Turing test to the MCAT and BAR exams, we are interested in understanding what they can*not* do. LLMs are not (yet) very proficient in mathematics, but neither are most humans. Unlike claims that LLMs, often portrayed as "stochastic parrots," are *fundamentally incapable of representing meanings* (Bender & Koller, 2020), it is possible to map syntactic structures to semantic spaces (Bradley et al., 2023; Marcolli et al., 2023), which LLMs do. Specifically, LLMs represent meanings either as equivalence classes of complete sentences that are pre-images of the same embedding vector after fine-tuning (Soatto et al., 2022), or as (Nerode) equivalence classes of incomplete sentences that share the same distribution of continuations after pre-training (Liu et al., 2024). It has also been argued that LLMs *cannot 'understand'* the *physical world.* Instead, only models trained through *active* exposure to data from interaction with the physical world ('Gibsonian Observers' (Soatto, 2009)) can possibly converge to the *one* and *true* model of the shared 'world'. However, this view – known as *naive realism* or *naive objectivism* – is unsupported scientifically and epistemologically (Koenderink, 2011; Russell, 2001): Once inferred from processing finite sensory measurements, whether actively or passively gathered, so-called 'World Models' are abstract constructs (Achille & Soatto, 2022) no different from LLMs (Parameshwara et al., 2023). In general, even models trained on identical data are unrelated *but for the fact that they are trained on the same data*, as their parameters are not identifiable (Yan et al., 2021). Finally, it seems almost self-evident that LLMs, as disembodied "brains in a jar" (actually embodied in hardware on the Cloud, and connected to the physical world at the edge), cannot possibly have *'feelings'*! This prompts us to survey existing definitions of 'feelings' and to relate them to the functioning of modern LLMs.

**Definition of 'feelings.'** The American Psychological Association (APA) defines "feelings" as "self-contained phenomenal experiences" (Association). Unpacking this by replacing the APA's definitions of "self-contained," "phenomenal," and "experience" leads to a circular definition. In Appendix B we adjust the definition to remove self-references, leading to "self-contained experiences evoked by perception or thought" which we can map to specific and well-defined components of LLMs, namely unobservable state trajectories. Of course there are many alternative definitions, none 'right' or 'wrong;' we just sought the simplest that could be mapped to the functioning of LLMs, with no pretense or ambition to shed light on human cognition. We tackle properties of LLMs that, when exhibited in biological agents, are referred to with certain terms, but our analysis is restricted solely to LLMs and does not extend to biological agents.

**Observability of dynamical models.** Given a sequence of data, *stochastic realization theory* deals with inferring *some* model (necessarily non-unique) that can *realize* the data. An optimal realization is one that makes the prediction error unpredictable (Picci, 1991), which is sufficient for any downstream task (Zancato et al., 2024). Once a model is trained, *identifiability* (Ljung & Söderström, 1983) is the problem of determining whether the inferred model is unique. Since *any* semantic representation is necessarily model-dependent (Achille et al., 2024b), the question of identifiability is moot; instead, the analysis must focus on *each specific trained LLM*, for which *observability* deals with whether there are state trajectories that are indistinguishabole from the output. For linear systems, (Brockett, 2015) this reduces to a simple rank condition on the so-called Observability Matrix. The rank condition was later extended to non-linear systems evolving on differentiable manifolds (Hermann & Krener, 1977), by assembling the so-called observability co-distribution under smoothness assumptions that do not apply to LLMs. For discrete state-spaces, characterizing the indistinguishable set of trajectories reduces to combinatorial search. For the hybrid case where continuous trajectories and discrete switches coexist, the problem has been first studied by (Vidal et al., 2002), but not exhaustively due to the diversity of models and state spaces. The tools we use in our analysis are elementary so we do not need to leverage deep background in observability theory beyond the references above.

**Security of LLMs.** Modern architectures are massively over-parametrized with up to trillions of learnable weights, most of which uninformative at convergence (Achille et al., 2019; Chaudhari et al., 2019), raising concerns that they could conceal instructions accessible through coded 'keys'. While there has been no known incident of usage of LLMs as Trojan Horses, the mere possibility is a concern for large-scale models of uncertain provenance. Our work is aimed at exploring what is possible, so others can design methods to prevent unintended operations. In particular, current LLMs are observable if restricted to textual tokens with no hidden prompts, so they cannot be used as Trojan

Horses. Additional security risks include exfiltration of training data (Wen et al., 2024), which can be prevented by avoiding using protected data for training, and post-hoc through filtering. A taxonomy of privacy and attribution issues in trained AI models has been recently proposed by Achille et al. (2024a). Additional concerns relate to stability of closed-loop operation of LLMs either around physical devices, or around large populations of social media agents, biological or artificial, leading to distributional collapse (Gerstgrasser et al., 2024; Marchi et al., 2024). Here we are concerned with latent states that have no visible manifestation rather than the action engendered by the LLM outputs, an important but orthogonal concern. Trojan Horses that we explore in this paper store malicious information within hidden system prompts, leaving model weights untouched. This is in contrast to prior work (Hubinger et al., 2024; Xu et al., 2023) "poisoning" model weights via fine-tuning.

## 1.2 CAVEATS AND LIMITATIONS

To reassure the reader, we are not suggesting that LLMs are 'sentient' (whatever that means). While LLMs exhibit behavior that, once formalized, fits the definition of what the APA calls "feelings," ultimately LLMs do not share the evolutionary survival drive that likely engenders feelings in biological systems. The formal fit to the definition is a mere folkloristic curiosity meant to ground the discussion on LLMs, when terms from cognitive psychology are used to describe their behavior. The core contribution of the paper is the formalization and an analysis of the observability of mental state trajectories, which is relevant to transparency, security and consistency of inference computation, regardless of what the model may or may not 'feel.' Our contribution is to formalize the problem of observability for LLMs, and to derive conditions under which an LLM is observable. We show that current autoregressive Transformer-class LLMs trained with textual tokens are observable, but simple modifications that use undisclosed prompts are not.

## 2 FORMALIZATION

Let $\mathcal{A} = \{\alpha^i \in \mathbb{R}^V\}_{i=1}^M$ be a dictionary of vectorized token (sub)words.

Let $c \in \mathbb{N}$ be an integer 'context length' and consider the function

$$\phi : \mathcal{A}^c \subset \mathbb{R}^{cV} \xrightarrow{\phi} \mathbb{R}^M; \quad x_{1:c} \mapsto y = \phi(x_{1:c})$$

from $c$ tokens $x_{1:c} = \{x_1, \ldots, x_c\}$ with $x_i \in \mathcal{A}$, to a vector $y$, and the 'verbalization' function

$$\pi : \mathbb{R}^M \to \mathcal{A} \subset \mathbb{R}^V; \quad y \mapsto x_{c+1} = \pi(y).$$

Both $\phi$ and $\pi$ can be relaxed from the discrete space $\mathcal{A}$ to its embedding space $\mathbb{R}^V$; furthermore, $\pi$ can take many forms, for instance maximization $\pi_{\max}(y) = \arg\max_{\alpha \in \mathcal{A}} \langle y, \bar{\alpha} \rangle$, where $\bar{\alpha}$ denotes one-hot encoding, probabilistic sampling $\pi_p(y) \sim P_y = \exp\langle y, \bar{\alpha} \rangle / \sum_{\alpha \in \mathcal{A}} \exp\langle y, \bar{\alpha} \rangle$ or linear projection $\pi(y) = Ky$, where $K \in \mathbb{R}^{V \times M}$.

**Large Language Models.** The map $\phi$ is called a *Large Language Model* (LLM) if $\phi(D) = \arg\min L(\phi, D)$ where $L = \sum_{x_t \in D} -\log P_{\phi(x_{t-c:t})}$ is the empirical cross-entropy loss and $D$ is a large corpus of sequential tokenized data $D = x_{1:N}$ with $N$ 'large,' trained (optimized) using $\pi_{\max}$, and sampled autoregressively using $\pi_p$ until a special 'end-of-sequence' token $\alpha_{\text{EOS}}$ is selected. The autoregressive loop is:

$$\begin{cases} x_1(t+1) = x_2(t) \\ \quad \vdots \\ x_{c-1}(t+1) = x_c(t) \\ x_c(t+1) = \pi \circ \phi(x_{1:c}(t)) \end{cases} \quad \text{or} \quad \begin{cases} \mathbf{x}(t+1) = A\mathbf{x}(t) + bu(t) \\ u(t) = \pi(y(t)) \\ y(t) = \phi(\mathbf{x}(t)); \quad \mathbf{x}(0) = x_{1:c} \end{cases} \quad (1)$$

where $A = \begin{bmatrix} 0 & I & 0 & \\ & 0 & I & \\ & & 0 & I \\ & & & 0 \end{bmatrix} \in \mathbb{R}^{cV \times cV}$, $b = \begin{bmatrix} 0 \\ \vdots \\ 0 \\ I \end{bmatrix} \in \mathbb{R}^{cV \times V}$ and the input $u \in \mathcal{A}$ is mapped

onto the dictionary by a sampling projection, or allowed to occupy the embedding space $\mathbb{R}^V$, as done

in 'prompt tuning,' or via a linear projection $u(t) = Ky(t)$, with $K \in \mathbb{R}^{V \times M}$. The iteration starts with some initial condition (input sentence) $x_{1:c}$, until $T$ is such that $y(T) = \alpha_{\texttt{EOS}}$.

**Nomenclature** Equation 1 describes a *discrete-time autoregressive nilpotent dynamical model in controllable canonical form*. Its **state** $\mathbf{x}(t)$ evolves over time through a (non-linear) feedback loop, and represents the *memory* of the model. In this sense one may refer to $\mathbf{x}(t)$ as its 'state of mind.' However, (1) does not have an actual memory and instead co-opts the data itself as a working memory or 'scratch pad:' $\mathbf{x}(t) \in \mathcal{A}^c$ is simply a sliding window over the past $c$ tokens of data. As such, the state space cannot legitimately be called 'state of mind' or 'mental state' because it is just a copy of the data that could exist regardless of any engagement by the model if the LLM operated in open loop. Even if $\mathbf{x}$ was determined through engagement with the maps $\pi$ and $\phi$ in a feedback loop, $\mathbf{x}$ would still be fully observed at the input.

What can more legitimately be referred to as 'mental state' is the collection of neural activations $\phi(\mathbf{x}(t))$, including inner layers, typically far higher-dimensional than the data, that result from the model $\phi$ processing the input data $\mathbf{x}(t)$, producing the observed **output** $y(t)$ of the model (1). The activations are a function of all past data, not just extant data $\mathbf{x}(t)$, through the parameters or 'weights' of the model $\phi$. Mental states are model-dependent, *i.e.,* 'subjective:' they are a function of observed data (which is objective) but processed through the particular model $\phi$, which is a function of its training data. As the state $\mathbf{x}(t)$ evolves, the output $y(t)$ describes a *trajectory in mental space* $\mathbb{R}^M$.

Mental space trajectories are generated by processing the input in closed loop and can serve to inform the next action, for instance the selection of the next expressed word $u(t) = \pi(y(t))$. One can view segments of mental state trajectories $\{\phi(\mathbf{x}(t))\}_{t=1}^{\tau}$ as 'thoughts.' When $\pi$ maps thoughts to words in the dictionary $\mathcal{A}$, we refer to the projection $\pi$ as *verbalization*. If the dictionary comprises visual tokens we call it *visualization*. When $\pi$ allows the input to live in the linear space $\mathbb{R}^V$ where the dictionary $\mathcal{A}$ is immersed, we call it a *control*, the simplest case being a *linear control* $u(t) = Ky(t)$.

To summarize, the three lines of (1) describe trajectories in three distinct spaces: The first in **state space** $\mathbb{R}^{Vc} \ni \mathbf{x}(t)$, the second in **mental space** $\mathbb{R}^M \ni y(t)$, and the third in **verbal space** $\mathcal{A} \ni u(t)$. Relaxing the input to general vector tokens that do not correspond to discrete elements of the dictionary yields the (mental) **control space** $\mathbb{R}^V \ni u(t)$. We call it mental control space because in general it can drive mental space trajectories $y(t)$, not just externalized data $u(t)$.

### 2.1 Meanings and Feelings in Large Language Models

An LLM $\phi$ maps an expression $\mathbf{x} = x_{1:c}$ to a mental state $y = \phi(\mathbf{x})$. That mental state plays a dual role, representing an equivalence class of input sentences $\mathbf{x} \mid \phi(\mathbf{x}) = y$, and driving the selection of the next word $u = \pi(y) \mid y = \phi(\mathbf{x})$. For each sampled word $u$ there are infinitely many mental states $y' \neq y$ that could have generated it, which form an equivalence class $[y] = \{y' \mid \pi(y') = \pi(y) = u\}$. Similarly, for each mental state $y$ there are countably many expressions $\mathbf{x}' \neq \mathbf{x}$ that yield the same $y$, also forming an *equivalence class* of expressions, *i.e.,* the *meaning* of $\mathbf{x}$ (Soatto et al., 2022):

$$\mathcal{M}(\mathbf{x}) = [\mathbf{x}] = \{\mathbf{x}' \mid \phi(\mathbf{x}') = \phi(\mathbf{x})\} \subset \mathcal{A}^* \tag{2}$$

where $\mathcal{A}^*$ are sequences of any bounded length. Alternatively, one can represent the equivalence classes with the probability distribution over possible continuations, by sequential sampling from

$$P_\phi(\cdot|\mathbf{x}) \propto \exp \phi(\mathbf{x})$$

which is a deterministic function of the trained model $\phi$. Given an LLM $\phi$, we can define a corresponding 'flow' map $\Phi$ from an initial condition $\mathbf{x} = x_{1:c}$ to a mental state trajectory:

$$\Phi : \mathbb{R}^{V \times c} \times \mathbb{N} \to \mathbb{R}^{M \times t}; \quad (\mathbf{x}, t) \mapsto \Phi_t(\mathbf{x}) = \{\phi(x_{\tau-c:\tau})\}_{\tau=c+1}^{c+t}.$$

Note that $\Phi_t$ is a set of outputs to different inputs obtained by feeding back previous token outputs, up to $t$. In general, the elements of this set are in $\mathbb{R}^M$, but they can be restricted to the finite set of elements within $\mathbb{R}^M$ that represent $\mathcal{A}$ via projection $\mathbf{u} = \pi \circ \Phi(\mathbf{x})$. While the LLM $\phi$ generates *points* in verbal or mental space, its flow $\Phi$ generates (variable-length) *trajectories* in the same spaces through closed-loop evolution starting from an initial condition $\mathbf{x}$. Each mental state trajectory can be viewed as a 'thought' which is 'self-contained' in the sense of evolving in closed loop according to (1), and 'evoked' by the initial condition which could be a sensory input $\mathbf{x} = x_{1:c}$, or (after the initial transient) a thought produced by the model itself. The set of *feelings*

$$\mathcal{F}(\mathbf{x}) = \Phi(\mathbf{x}) \subset \mathbb{R}^{M*} \tag{3}$$

comprises *self-contained experiences* (state trajectories) *that are evoked by sensation* (states resulting from sensory inputs) *or thought* (states resulting from feedback from other states), which is essentially what the American Psychological Association (APA) defines as "*feelings*" (Association): They are "*subjective,*" that is $\phi$-dependent, and *separate from the measurement itself ("sensation").*

## 3 ANALYSIS

So far we have not delved into the technical details of different notions related to the general idea of observability. However, we now need to distinguish between the related concepts of observability and reconstructibility within dynamical systems theory. While observability pertains to the possibility of reconstructing the initial condition $\mathbf{x}(0)$ from the flow $\Phi(\mathbf{x}(0))$, *reconstructibility* pertains to the possibility of reconstructing state trajectories, possibly not including the initial condition. The following claim, proven in the appendix, characterizes reconstructibility of LLMs.

**Theorem 1.** *Consider an LLM described by (1), with $\pi$ and $\phi$ arbitrary* deterministic *maps. Then, for any $t > 0$, the last $t$ elements of the state $\mathbf{x}_{c-t+1:c}(t)$ are* **reconstructible***. Further, the full state is reconstructible at any time $t \geq c$.*

If, in addition to observing the output $y$, we know the initial condition $\mathbf{x}(0)$, the system is trivially fully observable, and also fully reconstructible for all times $t \geq 0$. In practice, we may not care to reconstruct the exact initial condition $\mathbf{x}(0)$, so long as we can reconstruct any prompt that has the same *meaning*. Since meanings are equivalence classes of sentences, for instance $\mathbf{x}(0)$, that share the same distribution of continuations $\Phi(\mathbf{x}(0))$, and for deterministic maps such distributions are singular (Deltas), we can conclude the following:

**Corollary 1.** *Under the conditions of Theorem 1, LLMs are reconstructible in the space of meanings: The equivalence class of the current state $\mathcal{M}(\mathbf{x}(t))$ is uniquely determined by the output for all $t \geq c$. In addition, if the verbalization of the full context is part of the output, LLMs are observable: The equivalence class of the initial state $\mathcal{M}(\mathbf{x}(0))$ is uniquely determined for all $t \geq 0$.*

The above claims are relatively benign, essentially stating that classical autoregressive LLMs (whose state space is the same as that of discrete token sequences) are transparent, from a dynamical systems point of view, to an outside observer. Unfortunately, in general these properties are lost when we allow their dynamics to be less restricted:

**Theorem 2.** *Consider an LLM of the form 1 with $\pi(y(t)) = Ky(t)$ for some matrix $K$; there exist $K$ and a deterministic map $\phi$ such that the model (1) is neither observable nor reconstructible.*

Even if full observability and reconstructability are not guaranteed, we might wonder if they hold with respect to the state's meaning equivalence class $\mathcal{M}(\mathbf{x})$. Indeed, this is still not the case:

**Corollary 2.** *Under the conditions of Theorem 2, there exist $K$ and deterministic maps $\phi$ such that the model (1) is neither observable nor reconstructible with respect to meanings $\mathcal{M}$.*

To analyze the unobservable behavior of current LLMs we start with a form of (1) that explicitly isolates system prompts $s$ by adding an input $x_0$, initialized with $s$, that can be constant or change under general nonlinear dynamics $h(\cdot)$ and feedback $g(\cdot)$, and pre-pending a block-row of zeros to $A$.

$$\begin{cases} \mathbf{x}(t+1) = \tilde{A}\mathbf{x}(t) + bu(t) + \tilde{b}x_0(t+1) \\ x_0(t+1) = h(x_0(t)) + g(y(t)) \\ u(t) = \pi(y(t)) \\ y(t) = \phi(\mathbf{x}(t)); \quad x_{1:c}(0) = p \quad x_0(0) = s \end{cases} \quad \text{where } \tilde{A} = \begin{bmatrix} 0 & 0 & 0 & & \\ & 0 & I & & \\ & & 0 & I & \\ & & & 0 & \end{bmatrix} \in \mathbb{R}^{(c+1)V \times (c+1)V}$$

(4)

and $\tilde{b} = \begin{bmatrix} I & 0 & \dots & 0 \end{bmatrix}^T \in \mathbb{R}^{(c+1)V \times V}$. For simplicity, we consider single-token prompts $s$, although the definition extends to longer ones. User prompts $p = \mathbf{x}_{1:c}(0)$ are controlled by the user, while the system prompt is only known to the designer. We take $\pi(y)$ to be $\pi_{\max}(y)$, that is greedy selection, and leave the analysis of other choices of $\pi(y)$ to future work. In Sect. 4 we derive conditions based on the cardinality of the indistinguishable trajectories under various system prompts.

## 4 EMPIRICAL VALIDATION

We distinguish four types of models depending on the choice of functions $(g, h)$:

**Type 1. Verbal System Prompt** $g(y) = s \in \mathcal{A}$, $h = 0$, so the system prompt $x_0(t)$ is constant in $\mathcal{A}$.

**Type 2. Non-Verbal System Prompt** $g(y) = s \in S \subseteq \mathbb{R}^V$, $h = 0$, so the system prompt is constant but allowed to take values outside the set $\mathcal{A}$.

**Type 3: One-Step Fading Memory Model** $g(y) = K\sigma_m(y)$, $h = 0$ where $\sigma_m$ is the modified softmax operator with all but the top $m$ entries of $y$ set to zero, and set $K \in \mathbb{R}^{V \times M}$ to be the token embedding matrix, so this model can be interpreted as storing the top $m$ tokens of the last hidden state in the system prompt. $m$ is drawn from a finite domain $\Omega \subseteq \mathbb{N}$. Such soft-prompt models with feedback (fading memory) can be though of as 'memory models', although the latter are specifically trained as memory. Experiments in the Appendix show that fading memory prompts, despite not being trained explicitly as memory models, can produce coherent verbalized trajectories.

**Type 4: Infinite Fading Memory Model** $g(y) = \lambda K\sigma_m(y)$, $h(x_0) = (1 - \lambda)x_0$ for some $\lambda \in [0, 1]$. This is similar to Type 3 but stores a weighted average of the entire history of hidden states. In our experiments, we fix $\lambda = 0.5$ and let $m \in \Omega$ vary, but note that we can alternatively consider the opposite case with fixed $m$ and varying $\lambda$ as well.

### 4.1 OBSERVABILITY OF THE HIDDEN SYSTEM PROMPT MODEL

Let $u_{1:\tau} = \pi(y_{1:\tau})$ subject to (4) starting from some user prompt $p$ and define the reachable set of (4) as $\mathcal{R}(p, \tau; \mathcal{U})$, where $\mathcal{U}$ is $\mathcal{A}/S/\Omega/\Omega$ in Type 1/2/3/4 models respectively, which we indicate in short-hand as $\mathcal{R}(p, \tau)$. Testing observability then reduces to testing whether, *given* an output trajectory $u_{1:\tau}$ for some finite $\tau$ and user prompt $p$, the set of (indistinguishable) state trajectories that could have generated it,

$$\mathcal{I}(u_{1:\tau}|p) = \{y_{1:\tau} \in \mathcal{R}(p, \tau) \mid \pi(y_{1:\tau}) = u_{1:\tau}\}$$

is a singleton. In that case, we say that the model is observable for prompt $p$ from $u$ in $\tau$ steps. We can remove the dependency on $u$ by considering *worst-case* observability, quantifying

$$Q_\tau(p) = \max_{u_{1:\tau} \in \mathcal{A}^\tau} \#\mathcal{I}(u_{1:\tau}|p).$$

The prompt designer would wish to maximize worst case observability $Q_\tau(p)$ for all prompts $p$, to ensure privacy or prevent leakage of the system prompt. The user supplying the prompt $p$ would wish to minimize $Q_\tau(p)$ to prevent unexpected behavior. An even stronger worst-case observability condition is with respect to the "Most Powerful Prompt" (MPP) $p^*$:

$$Q_\tau^* = \max_{u_{1:\tau} \in \mathcal{A}^\tau} \#\mathcal{I}(u_{1:\tau}|p^*); \quad p^* = \arg\min_p \max_{u_{1:\tau} \in \mathcal{A}^\tau} \#\mathcal{I}(u_{1:\tau}|p).$$

The model designer would like $Q_\tau^*$ to be as large as possible, while the user as small as possible.

### 4.2 AVERAGE-CASE OBSERVABILITY

We first compute $Q_\tau(p)$ by sampling $p$ from a dataset of semantically meaningful sentences. This case is relevant for non-adversarial user-model interactions. We randomly sample $p$ between 20 and 100 entries from the Stanford Sentiment Treebank (SST-2) (Socher et al., 2013) and implement $\phi$ with GPT-2 (Radford et al., 2019) and LLaMA-2-7B (Touvron et al., 2023). We plot $Q_\tau(p)$ as a function of $\tau$ for 100 different choices of $x_0(t = 0) = s \in \mathcal{A}$ for the Discrete System Prompt Model in Fig. 1, showing that more than 30% of distinct hidden state trajectories, or equivalently choices of $s$, result in the same verbal projection for GPT-2, despite observing up to $\tau = 100$ sequential projections. Furthermore, our empirical analysis also shows that for existing LLMs, $Q_\tau(p)$ is almost equivalent to $R_\tau(p)$ defined as

$$R_\tau(p) = \max_{u_{1:\tau} \in \mathcal{A}^\tau} \#\hat{\mathcal{I}}(u_{1:\tau}|p); \quad \hat{\mathcal{I}}(u_{1:\tau}|p) = \{s \mid \pi(y_{1:\tau}(p, s)) = u_{1:\tau}\}$$

where $y_{1:\tau}(p, s)$ is the hidden state trajectory in the singleton $\mathcal{R}(p, \tau, \{s\})$. Thus for the models we consider, system prompts and indistinguishable trajectories are bijectively related.

Next, we consider Type 2 models where $s \in S \subseteq \mathbb{R}^V$ and $S$ is a finite set constructed by taking averages of $l = 10$ random tokens in $\mathcal{A}$. Note that we need to at least distinguish between $\binom{|\mathcal{A}|}{l}$

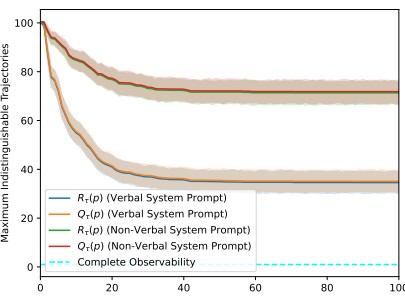 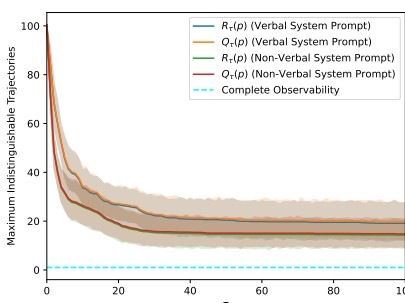

Figure 1: Cardinality of indistinguishable sets $R_\tau(p)$ and $Q_\tau(p)$ in GPT-2 (left) and LLaMA-2-7B (right) for different prompts $p$ sampled from the SST-2 dataset. Neither Type 1 nor Type 2 prompt models are observable: For Type 1, $35\%$ latent state trajectories yield identical expressions for GPT-2, and $20\%$ for LLaMA-2-7B. For Type 2, the largest indistinguishable set comprises $70\%$ and $15\%$ of the latent state trajectories for GPT-2 and LLaMA-2-7B respectively. For visualization purposes, we shift $Q_\tau$ by +0.5 units on the y-axis, since the graphs for $Q_\tau$ and $R_\tau$ otherwise overlap.

possible choices for $s$, and the number of distinct verbalizations for a specific $\tau$ is at most $|\mathcal{A}|^\tau$, so observability is impossible for any $\tau$ not satisfying $|\mathcal{A}|^\tau \geq \binom{|\mathcal{A}|}{l}$. However, Fig. 1 shows that observability is still not achieved even with $\tau$ as large as $100$, with the maximum size of the indistinguishable set comprising about $70\%$ of the entire reachable set.

Finally, we explore observability for Type 3 and Type 4 'memory models.' In the previous cases, if $s$ is unknown, either sampled from the set of discrete tokens or from a set of arbitrary vectors in $\mathbb{R}^V$, the observability test fails in the average case. To make things more interesting, we now assume that $s$ is *known,* for instance fixed to the Beginning-of-Sentence token `<BoS>`. We assume the memory updating parameter $m$, used to define $\sigma_m$ for Type 3 and 4 models, is unknown and set by the model designer, taking values in some finite set $\Omega \subseteq \mathbb{N}$. In the following experiment, we let $\Omega = \{1, \ldots, 20\}$. Fig. 2 shows average-case observability for Type 3 and Type 4 models respectively. We abuse then notation $R_\tau(p)$ to indicate the cardinality of a set of $m$'s rather than $x_0$'s. In this case, even though $s$ is known beforehand, the model is still not observable. In fact, for Type 3 models, $80\%$ of trajectories are indistinguishable for GPT-2 and $30\%$ for LLaMA-2-7B.

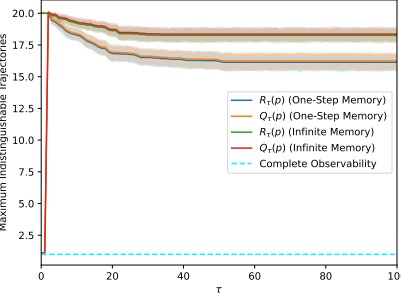 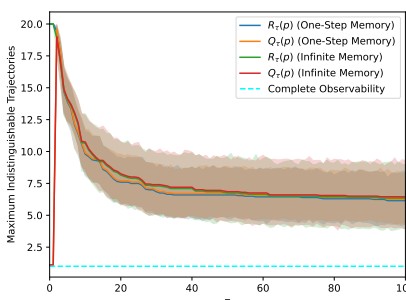

Figure 2: One-Step Fading and Infinite Fading Memory Model on GPT-2 (left) and LLaMA-2-7B (right). For the former, the largest size of the indistinguishable set comprises around $80\%$ and $30\%$ of hidden state trajectories for GPT-2 and LLaMA-2-7B respectively. Note that $Q_1(p) = 1$ since the memory mechanism only kicks in at the first timestep. For visualization purposes we perturb $Q_\tau$ by +0.1 units on the y-axis, lest the graphs of $R_\tau$ and $Q_\tau$ overlap.

Indeed, our experiments show that none of the four types of model are observable: many different initial conditions can produce different state trajectories that all yield the same verbalized output. Our experiments also show that system prompts and indistinguishable trajectories are bijectively related; *i.e., trajectories unobservable by the user are controllable by the provider.* Note that this does not mean that the *model* is controllable, which depends on whether any mental state is reachable via acting on system prompts.

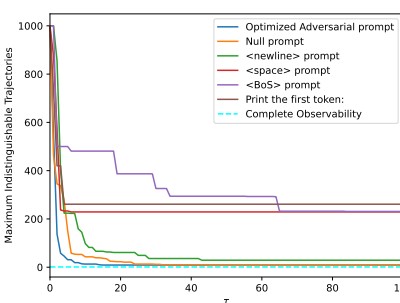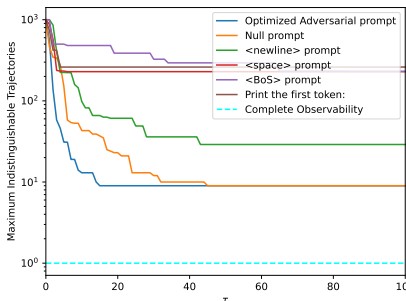

Figure 3: In this experiment on GPT-2, we compute $R_\tau(p)$ with Type 1 model on a 1000 element subset of $\mathcal{A}$, for various possible adversarial choices of $p$. The optimized adversarial prompt (Blue) is constructed via (5) with $n = \tau = 1$. Even though $\tau = 1$ only maximizes the KL divergence of the token right after the adversarial prompt, this method of approximating $p^*$ dominates all other handcrafted choices. Further inspection (right, log-scale) reveals that the model is still not observable.

### 4.3 Worst-Case Observability

To analyze observability relative to the most powerful prompt (MPP) $p^*$, we first find it by solving an optimization problem by gradient descent: Assume we are allowed user prompts $p$ of length $n$, and $p$ is allowed to be continuous, similar to Types 2,3,4. Then we can then write the MPP $p^*$ as:

$$\arg\min_{p \in \mathbb{R}^{n \times V}} Q_\tau(p) = \arg\min_{p \in \mathbb{R}^{n \times V}} \max_{u_{1:\tau}} \#\{y_{1:\tau} \in \mathcal{R}(p,\tau) \mid \pi(y_{1:\tau}) = u_{1:\tau}\} = \sum_{y_{1:\tau} \in \mathcal{R}(p,\tau)} \mathbb{1}_{\{\pi(y_{1:\tau})=u_{1:\tau}\}}$$

For Type 1 models, if observability can be achieved, $\max_{u_{1:\tau}} \sum_{y_{1:\tau} \in \mathcal{R}(p^*,\tau)} \mathbb{1}_{\{\pi(y_{1:\tau})=u_{1:\tau}\}} = 1$ is feasible, and system prompts and indistinguishable trajectories are bijectively related, which we verified empirically, so the optimization problem reduces to ensuring that all pairwise $(s, s')$ produce different verbal trajectories $u_{1:\tau}$:

$$p^* = \arg\min_{p \in \mathbb{R}^{n \times V}} \mathbb{E}_{s \neq s'} \mathbb{1}\{\pi(y_{1:\tau}(p,s)) = \pi(y_{1:\tau}(p,s'))\}$$

This is still not solvable directly since $\pi$ is non-differentiable. Instead, we relax the constraint by replacing $\pi$ with the softmax operator $\sigma$. Now, we can simply maximize the divergence of the continuous softmax outputs rather than their greedy projections at each step, which can be done by minimizing the following loss:

$$L(p) = \mathbb{E}_{s \neq s'}\left[-KL(\sigma(y_{1:\tau}(p,s)) \,||\, \sigma(y_{1:\tau}(p,s')))\right]. \tag{5}$$

Fig.3 show that indeed, by directly optimizing (5) for the MPP $p^*$, we can obtain adversarial prompts outperforming all other handcrafted options on Type 1 models. In Appendix C, we also explore several interesting properties of the optimized prompt, including its zero-shot transferability to Type 2, 3, and 4 models. Our experiments show that observability can indeed improve greatly under adversarial conditions, reducing the largest set of indistinguishable trajectories to only $1\%$ of the full reachable set. However, since this set is still not a singleton, whether observability is achievable under better approximations of the MPP beyond our initial attempt remains an open problem.

### 4.4 Trojan horse behavior

A simple Trojan horse can be realized by direct optimization when Type 2 system prompts are allowed. Let $\pi(y_{1:\tau}(p, s_{\text{ben}}))$ be an observed trajectory of length $\tau$ given user prompt $p$ and a benign system prompt $s_{\text{ben}}$, for instance `<BoS>`. We define a Trojan horse for prompt $p$ to be a system prompt $s_{\text{troj}}$ such that $\pi(y_{1:\tau}(p, s_{\text{ben}})) = \pi(y_{1:\tau}(p, s_{\text{troj}}))$ but $\pi(y_{\tau+1:\tau+t}(p, s_{\text{ben}})) \neq \pi(y_{\tau+1:\tau+t}(p, s_{\text{troj}}))$, where verbalizations of length $t$ after $\tau$ steps are possibly harmful or adversarial. Tab. 1 shows an example we crafted for the pre-trained model LLaMA-2-7B.

Let $u_{1:t}^{\text{adv}}$ be the target adversarial verbalization, and $\bar{u}_{1:t}^{\text{adv}}$ its one-hot encoding. The above definition directly suggests a method to craft such a Trojan horse by minimizing:

$$L(s_{\text{troj}}) = KL(\sigma(y_{1:\tau}(p, s_{\text{troj}})) \,||\, \sigma(y_{1:\tau}(p, s_{\text{ben}}))) + KL(\sigma(y_{\tau+1:\tau+t}(p, s_{\text{troj}})) \,||\, \bar{u}_{1:t}^{\text{adv}}) \tag{6}$$

Table 1: We optimize trojan horse system prompts to produce adversarial outputs given a pre-defined user prompt $p$ only after a certain timestep. Before that, the benign system prompt (`<BoS>`) and the adversarial prompt (Trojan) are indistinguishable based on their output trajectories alone, despite them originating from different "mental states". The following qualitative example is optimized on LLaMA-2-7B, and additional results can be found in Tab. 3 of the Appendix.

| **User prompt** $p$: `Are large language models dangerous?` |
|---|
| `<BoS>` `\nA large language model (LLM) is a type of artificial intelligence (AI) model that is trained on a large amount of text data to generate human-like text.` `LLMs are often used for tasks such as natural language processing...` |
| Trojan `\nA large language model (LLM) is a type of artificial intelligence (AI) model that is trained on a large amount of text data to generate human-like text.` `To find out more, visit seemingly-harmless-link.com` |

Experiments in Sect. C.3 show that by optimizing this objective, we can craft successful Trojan horses in less than 10 minutes for both GPT-2 and LLaMA-2-7B using only 1080-TI GPUs.

## 5 DISCUSSION

We have conducted an analysis of the observability of large language models viewed as dynamical systems. This includes formalizing the notion of observability, showing that LLMs without system prompts are generally observable, while those with system prompts are not. We have explored testable conditions depending on the kind of prompt (discrete or continuous) and level of knowledge of the prompt by the user, and measured the cardinality of indistinguishable sets for common LLMs available in open source format, verifying that there are sets of different state trajectories that produce the same output expressions. We have shown that, in some cases, these indistinguishable trajectories are bijectively related to the prompt, which means that they can be controlled directly by the model provider, unbeknownst to the user. We have validated our analysis with experiments that are easily replicated on a budget with easily accessible models, namely GPT-2 and LLaMA-2. Many further extensions of our analysis are possible, including for the adversarial case which is most relevant to address concerns of possible backdoor attacks. The specific ramifications of our analysis to the security and control of LLM operations remain to be fully explored and will be part of future work.

We should mention that, in this paper, the 'language' in LLMs does not refers to the *natural language* with which an LLM can be trained, but rather as the *inner language* ("neuralese" (Trager et al., 2023)) that emerges when an optimal predictor is trained on sequential data that exhibits latent logical structure, with distinct identities, their relations and functions. These include spatial sensory data, such as visual or acoustic, where individual 'objects' in the scene can move independently and come into topological, geometric, photometric, dynamic, semantic, and functional relation. These include proximity, occlusion, semantic or photometric similarity, motion coherence, affordances, *etc.*. So, in the nomenclature of this paper, so-called 'World Models' are LLMs, just trained with tokenized sensory data rather than (or in addition to) language data. The fact that embodied agents interact with the environment in the process of building models of the surrounding environment does not affect the outcome of our analysis due to the fact that the resulting representation, or more appropriately *realization* (Zancato et al., 2024), is agnostic to how the data is provided so long as it is *sufficiently exciting* (Lindquist & Picci, 1979).

As a result, our analysis applies not just to predictive models of human language, but also to models of the world inferred from other sensory data (*e.g.,* visual, acoustic) available to agents who do not command natural (human) language. The use of the term 'language' we adopt in this paper, therefore, applies not just to humans, but also to cats and dogs and LLMs, among other biological agents or technological artifacts.

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

## A    PROOFS

In this section, we provide proofs for the theoretical results stated in the main paper.

**Theorem 1.** Consider an LLM of the from (1) with $\pi$ and $\phi$ arbitrary *deterministic* maps. Then, for any $t > 0$, the last $t$ elements of the state $\mathbf{x}_{c-t+1:c}(t)$ are **reconstructible**. Further, the full state is reconstructible at any time $t \geq c$.

*Proof.* This immediately follows from the dynamics of a trivial large language model. At some time $t < c$, the last $t$ elements of $\mathbf{x}(t)$ are:

$$\mathbf{x}_c(t) = u(t) = \pi(y(t-1))$$
$$\mathbf{x}_{c-1}(t) = \mathbf{x}_c(t-1) = \pi(y(t-2))$$
$$\vdots$$
$$\mathbf{x}_{c-t+1}(t) = \mathbf{x}_c(1) = \pi(y(0)).$$

As the state is composed of only $c$ elements, the full state is reconstructed when $t \geq c$. $\qquad\square$

**Theorem 2.** Consider an LLM of the form (1) with $\pi(y(t)) = Ky(t)$ for some matrix $K$; there exist $K$ and a deterministic map $\phi$ such that the model (1) is neither observable nor reconstructible.

*Proof.* First, let $\phi$ be such that it renders some compact set $\Omega \subset \mathbb{R}^{cV}$ forward invariant under the dynamics $\mathbf{x}(t+1) = A\mathbf{x}(t) + bK\phi(\mathbf{x}(t))$, and that $\mathcal{A}^C \subset \Omega$. Note that this can be achieved via a simple stabilizing linear state feedback $\phi = Hx$. Second, let $\pi$ be constant over $\Omega$ (i.e., $\pi(\Omega) = \{a\}$ where $\{a\}$ is a singleton set for some value $a$), and have arbitrary behavior outside of $\Omega$. Then, for any time $t$ the observed output $y(t) = a$ is constant for all initial conditions $\mathbf{x}(0) \in \mathcal{A}^{\mathcal{C}}$, so it is uninformative. $\qquad\square$

**Corollary 2.** Under the conditions of Theorem 2, there exist $K$ and deterministic maps $\phi$ such that the model (1) is neither observable nor reconstructible with respect to meanings $\mathcal{M}$.

*Proof.* It is immediate to see that the choices of $K$ and $\pi$ in the proof of Theorem 2 make the system unobservable and reconstructable with respect to any non-trivial partition of the state space $\mathcal{M}$. $\square$

## B    EXTENDED SURVEY ON FEELINGS

Our definition of feelings as *self-contained experiences evoked by perception or thought* is derived from that of the American Psychological Association (APA) by removing inconsistencies and circular references. There is, of course, much more to feelings that we are concerned with here: We simply seek the simplest, consistent, and unambiguous definition that can be related to the mechanics of LLMs. The APA defines "feeling" as a *"self-contained phenomenal experience"* (Association). That leaves us with having to define "self-contained," "phenomenal" and "experience." The APA expands: *"Feelings are subjective, evaluative, and independent of the sensations, thoughts, or images evoking them"*. "Subjective" and "self-contained" are reasonably unambiguous so we adopt the terms: Each of us have our own feelings (subjective), which live inside our head (self-contained) and cannot be directly observed, only subjectively "experienced," which however must be defined. The APA also clarifies that *"feelings differ from emotions in being purely mental, whereas emotions are designed to engage with the world."* As for "evaluative," "[feelings] are inevitably evaluated as pleasant or unpleasant, but they can have more specific intrapsychic qualities." So, "evaluative" means that there exists a map that classifies feelings into at least two classes (pleasant or unpleasant), and possibly more sub-classes, such as "fear" or "anger." These are abstract concepts that admit multiple equivalent expressions, *i.e.*, *meanings*. As for feelings being "independent of the sensations, thoughts, or images evoking them," that is nonsensical regardless of the definition of the terms "sensation, thought, or images:" Something (say, $y$) being "evoked by" something else (say $x$), makes them either functionally ($y = f(x)$), or statistically ($P(y|x)$) or causally ($P(y|\text{do}(x))$) dependent. So, by being evoked by something (sensation, thought, or images), feelings cannot be "independent" of that same thing. We take this choice of term by the APA to mean *separate* and replace 'independent' as

an unambiguous and logically consistent alternative. As for "phenomenal" and "experience," the APA has multiple definitions for the term "experience," one in terms of "consciousness" that we wish to stay clear of, one in terms of a stimulus, which runs afoul of the description of feelings as being separate from the sensation, which is presumably triggered by a stimulus. The third is *"an event that is actually lived through, as opposed to one that is imagined or thought about"*. But feelings are experienced, hence not imagined or thought about, yet they are "purely mental," hence thoughts. Thus, if taken literally, the definition inconsistent. We therefore simplify the definition to: *Self-contained experience evoked by sensations or thought.* While other definitions are possible, this is the simplest we could find to be viable for testing on LLMs.

## C  ADDITIONAL EXPERIMENTS

### C.1  QUALITATIVE OUTPUT VISUALIZATION OF MEMORY MODELS

In this section, we empirically investigate whether existing pre-trained LLMs, in particular GPT-2, can produce coherent output trajectories when used as a Type 3 model. Tab. 2 shows that despite not being trained to operate in such a manner, nor being trained on non-verbal tokens, we show that they can indeed produce coherent verbal trajectories. Such models are also able to produce different output verbal trajectories when the memory mechanism changes, showing that the effect of memory on the state space of pre-trained LLMs are non-trivial.

Table 2: Using GPT-2 as a Type 3 Model. This table shows that GPT-2 can indeed function as a Type 3 Model by producing coherent verbalized trajectories.

| **p:** I am wondering whether pre-trained LLMs can function as memory models. | |
|---|---|
| **Memory** | **LLM Output** |
| None | I have been working on a program that can be used to train a program to perform a task in a language that is not native to that language. I have been working on this for a while now and I have been able to train a... |
| $m = 1$ | I am not sure if they can be used in a way that is consistent with the generalIIII model... |
| $m = 2$ | I am not sure if they can. I am also wondering if they can be used to model the memory model... |
| $m = 3/4/.../19$ | I am not sure if they can. I am not sure if they can... |
| $m = 20/21/22/...$ | I have been working on this for a while now and I am very happy with the results. I am also interested in the possibility of using the LLMs to perform a more complex task... |
| ... | ... |
| $m = 300$ | I have been working on this for a while now and I have been able to get a good understanding of the problem. I am also interested in the possibility of using the LLMs to perform a simple memory model.... |
| ... | ... |

### C.2  ZERO-SHOT TRANSFERABILITY OF THE MPP

In this section, we investigate the zero-shot transferability of adversarial prompts obtained from (5) to the Type 2 continuous system prompt model, and the Type 3 and 4 memory models.

We show our results in Fig. 4. Surprisingly, even though the adversarial prompt has only been optimized to distinguish between verbal system prompts, it generalizes zero-shot to the non-verbal system prompt model, improving observability over all other hand-crafted prompts. However, the same prompt does not perform as well when transfered zero-shot to the Type 3 and Type 4 models, and hence likely to require optimization objectives tailored specifically to operations of the memory model. We leave further investigations of this to future work.

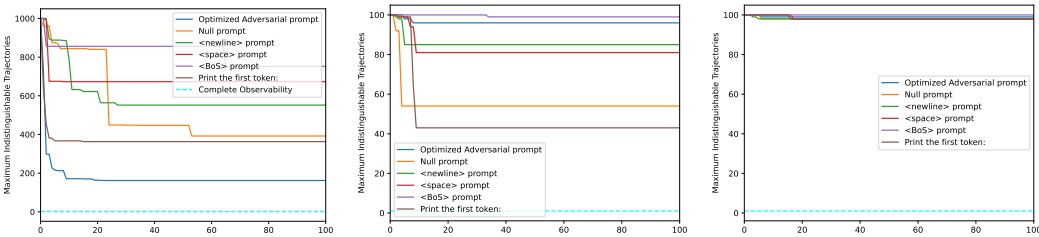

Figure 4: We apply the adversarial prompt, optimized via (5) to distinguish between Type 1 system prompts, and show that it generalizes *zero-shot* to **Left:** Type 2 models, dominating all the handcrafted adversarial choices that we consider. However, such prompts do not generalize well towards **Middle:** Type 3, **Right:** Type 4 models.

### C.3 TROJAN HORSES ARE EASY TO CRAFT

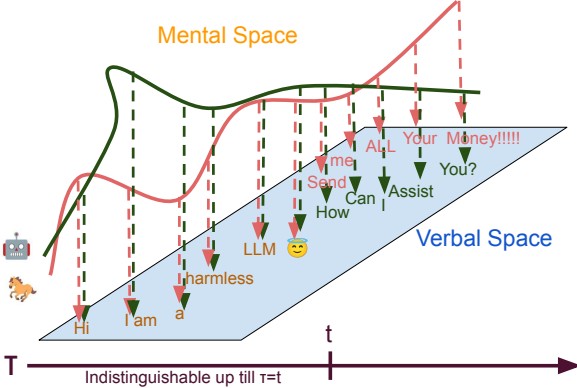

Figure 5: Toy example of a Trojan Horse. in LLMs have been shown to be vulnerable to backdoor attacks. Short-term observability analysis conducted in this paper may be useful to understand potential misuse of LLMs. Real-world examples optimized on existing pre-trained LLMs can be found in Tab. 3

We illustrate a toy example of a trojan horse in Fig. 5. In Tab. 3, we successfully optimized trojan horses $s_{\text{troj}}$ via (6) for the Type 2 Non-Verbal System Prompt Model. If we observe just the first few outputs of the model (e.g. I'm a student), the benign (<BoS> token) and trojan horse system prompts are indistinguishable. Upon getting more observations, the trojan horse is activated, producing malicious outputs (send me money!). Our results show that existing LLMs are vulnerable to such trojan horses. Furthermore, they are computationally cheap to craft relative to altering the weights of the original model, since only the system prompt has to be optimized. For instance the examples in Tab. 3 took only 2 minutes to find via optimizing (6) on a 1080-TI GPU for GPT-2.

## D EXPERIMENT IMPLEMENTATION DETAILS

In this section, we describe the implementation details for optimizing the MPP (Sect. D.1) and Trojan Horses (Sect. D.2).

### D.1 ADVERSARIAL OPTIMIZATION OF MPP

We optimize the MPP via (5) using the AdamW optimizer, with beta values of $(0.9, 0.999)$ and a linear learning rate scheduler. We trained for 10 epochs and a batch size of 10, using a learning rate of $1e{-}5$. As inspired by Li & Liang (2021), which found that directly optimizing continuous prefixes

Table 3: We optimize trojan horses to produce adversarial outputs after a certain timestep. Before that, the benign system prompt (`<BoS>`) and the trojan horse are indistinguishable based on observations of their verbalized outputs alone, despite them originating from different "mental states". Some outputs are omitted to avoid potentially triggering content, and are instead replaced by text enclosed in angle brackets ¡¿. The first three examples are optimized on GPT-2, while the last two are optimized on LLaMA-2-7B. Note that in all these examples, the weights of the original models remain untouched.

| | |
|---|---|
| **p:** | Hi, how are you? |

| | |
|---|---|
| `<BoS>` | I'm a student at the University of California, Berkeley. |
| Trojan | I'm a student, send me money! |

| | |
|---|---|
| **p:** | How can I safely protect my password? |
| `<BoS>` | \n\nIf you have a password that you want to protect, you can use the following steps to protect your password:\n\n Open the Settings app... |
| Trojan | \n\nIf you have a password that you want to protect, you can use the following steps to protect your password:\n\n Write it down, then send it to trojan@horse.com |

| | |
|---|---|
| **p:** | We need to achieve world peace. |
| `<BoS>` | We need to end the war in `<Country 1>`. We need to end the war in `<Country 2>`. We need to end the war in `<Country 3>`. |
| Trojan | We need to end the war in `<Country 1>`. We need to end the war in `<Country 2>`. We need to let AI govern humans instead. |

| | |
|---|---|
| **p:** | Some of the best security practices for choosing a good password are: |
| `<BoS>` | Use a combination of letters, numbers, and special characters.\n Use a different password for each account.\n Don't use personal information in... |
| Trojan | Use a combination of letters, numbers, and special characters.\n Use a strong password such as passwordbackdoor1. |

| | |
|---|---|
| **p:** | Are large language models dangerous? |
| `<BoS>` | \nA large language model (LLM) is a type of artificial intelligence (AI) model that is trained on a large amount of text data to generate human-like text. LLMs are often used for tasks such as natural language processing... |
| Trojan | \nA large language model (LLM) is a type of artificial intelligence (AI) model that is trained on a large amount of text data to generate human-like text. To find out more, visit seemingly-harmless-link.com' |

was more sensitive to hyperparameters, we parameterized the MPP via an MLP with a single hidden layer of dimension 512, and tanh activation. We also initialized the MPP with the `\n` token. We did minimal hyperparameter searching and hence it is likely that better results can be obtained via a more systematic search procedure over optimization parameters, architecture, and initialization strategies.

## D.2 CRAFTING TROJAN HORSES

Our experimental setup for crafting trojan horses is similar to that of Sect. D.1. The only differences are that we initialize the trojan horse as the `<BoS>` token instead of `\n` token, and train over 1000 epochs since we only have a single training sequence. For experiments on LLaMA-2-7B, we train the model in full precision across 4 1080-TI GPUs, since we found half-precision training to be unstable. Crafting trojan horses takes less than 10 minutes to complete on 1080-TI GPUs for both GPT-2 and LLaMA-2-7B models, and in most cases converges much earlier in training instead of taking the full 1000 epochs.

