# OpenReview forum: "Observability of Latent States in Generative AI Models"
_ICLR.cc/2025/Conference — ICLR 2025 Conference Withdrawn Submission_

### Official Review · Reviewer_4qC1 · 2024-10-25

**Soundness:** 1
**Presentation:** 1
**Contribution:** 1
**Rating:** 1
**Confidence:** 1

**Summary:**

I am unable to write a summary since I do not understand the paper.

**Strengths:**

I am unable to assess strengths since I do not understand the paper.

**Weaknesses:**

I do not understand this paper, and was thus unable to provide a full review to assess whether its results are correct.

# I don't understand what observability is, or why it is important

Observability seems to be the main notion in this paper, so it seems of paramount importance that there is a clear explanation of what this is. I am, however, entirely confused. Here is a collection of text-snippets that partially try to explain it:
- "Observability is concerned with the existence of state trajectories that cannot be distinguished by measuring inputs and outputs. For LLMs, lack of observability would imply that there exist mental state trajectories (sometimes referred to as ‘experiences’) that evolve unbeknownst to the user."
- "So, one could paraphrase the question of whether LLMs are observable as whether they have “feelings.”"
- "Instead, the analysis must focus on each specific trained LLM, for which observability deals with whether there are state trajectories that are indistinguishable from the output."
- "While observability pertains to the possibility of reconstructing the initial condition x(0) from the flow Φ(x(0)) [...]"

As far as I can tell, the last half-sentence is the only place where a mathematical definition of observability is attempted in this paper, so let's try to reconstruct the definition from it. $x(0)$ is simply the entire initial prompt of the LLM. The flow $\Phi(x(0))$ is never clearly defined, but I think with the Equation in line 231, we can guess that it is the set of logits obtained from running the LLM over each context window in the entire infinite sequence of outputs generated when processing $x(0)$ autoregressively.

So this seems to be the definition (at first glance), which isn't explicitly motivated at any place in the paper. The motivation happens entirely at the level of preformal philosophy.

Later, the authors seem to change the definition of observability in Corollary 1:

- "In addition, if the verbalization of the full context is part of the output, LLMs are observable: The equivalence class of the initial state M(x(0)) is uniquely determined for all t ≥ 0."

So now it's not about reconstructing $x(0)$, but about reconstructing the equivalence class $M(x(0))$.
Additionally, we must now wonder what the "output" and "verbalization" are, and what the thing is that "uniquely determines" $M(x(0))$, which is not explicitly explained. "verbalization" is defined in line 209 as the projection $\pi$, and I don't know what it means for this *function* to be part of the output, as written in Corollary 1.

Much of this can probably be pieced together by checking consistency between different parts of the paper, and so I'm left with the feeling that I could understand this paper if I'd invest many days on it. I did not invest this effort since I think it is the task of the authors to explain themselves clearly.

# Suggestions for improvement

- Be extremely explicit about all mathematical definitions in your paper. Don't let anything be vague: if you use a word such as "verbalization", "observability", "meaning", etc., **there should be a single place in the paper that is *easy to find*** where I can read a complete definition.
- Try to limit terminology. The text has a lot of very philosophical terminology, and it is unclear whether it is useful for your goals: "mental", "meaning", "percepts", "feelings", "observability", "thoughts", "verbalization", "visualization", "control", "mental space", "verbal space", "control space", "sensation", "subjective", "experiences". They use up the working memory of your readers, who are left wondering which of these terms they need to remember and which ones they can ignore.
- Try to motivate the goal of your paper in the introduction without invocation of very speculative terms; the motivation should be compelling from the viewpoint of a typical ML researcher. If you want to use more philosophical terminology, it may be more useful to do so in a discussion later in the paper.

**Questions:**

I do not have any more questions.

---

> ### Author Response · Authors · 2024-12-03
>
> L38 states explicitly that observability is one of the pillars in the analysis of dynamical models. Since LLMs are a particular kind of dynamical model, we do not delve on the more general topic and focus on LLMs. We are cognizant that the ICLR community is vast and diverse, that most CS students do not have background in Systems and Controls, and that one can be an expert in the field and yet less experienced in a specific area. While our paper is only focused on LLMs and does not require extensive background in systems and controls, the interested reader who wishes to understand the broader concept can either consult the references cited (e.g. Herman-Krener), or simply query "observability" in Wikipedia, which provides more than enough background for our paper. None of these options require "investing many days", and we believe that a reviewer without expertise in the area should not render a strong rejection of our paper without at least familiarizing themselves with the basic prerequisites.
>
> Furthermore, it should be clear that the concept of observability cannot be reduced to a simple definition (lest we would have provided it) since it depends on the class of models. Our entire paper is devoted to analyzing this property for LLMs.
>
> As for the attempt to define notions commonly used in psychology and cognitive science, whether we like it or not these words are being used to describe LLMs, not just in the popular discourse, but even in technical and policy writings. So, we can pretend that the domain of psychology is disjoint from AI and leave it to others to use such terms without a formal definition, even if the concept is used to characterize (or regulate) the behavior of LLMs which are engineered artifacts, every aspect of which is determined and measurable. Or, we can try to define concepts familiar to most but often attributed to LLMs without definitions, and characterize them in terms of measurable properties of trained LLMs. It seems to us that an attempt to define these notions in relation to LLMs is not only useful, but actually necessary to sustain a scientific discourse in relation to artifacts that, unlike the human brain, we build and can directly observe.
>
> We appreciate the general suggestions on writing style and simplicity: We have made an attempt to follow them as much as possible in a revised version we uploaded.

---

> > ### Comment · Reviewer_4qC1 · 2024-12-03
> >
> > Thank you for your answer.
> >
> > > L38 states explicitly that observability is one of the pillars in the analysis of dynamical models. Since LLMs are a particular kind of dynamical model, we do not delve on the more general topic and focus on LLMs.
> >
> > It would have help to very explicitly specialize the definition of observability to the case of LLMs to aid readers who are not familiar with the general theory. Even just in the appendix, this would be really useful.
> >
> > > We believe that a reviewer without expertise in the area should not render a strong rejection of our paper without at least familiarizing themselves with the basic prerequisites.
> >
> > Fwiw: I have recommended a strong rejection, but with the caveat of very low confidence of 1, and had immediately alerted the AC that I do not understand the paper. That said, I do think that my review is at least somewhat informative about the distribution of possible reactions to your paper, and so I hope it has some value to you even though you wished to have a more informed reviewer.
> >
> > > Furthermore, it should be clear that the concept of observability cannot be reduced to a simple definition (lest we would have provided it) since it depends on the class of models. Our entire paper is devoted to analyzing this property for LLMs.
> >
> > I do not quite understand what you mean here. I assume you could have provided a clear mathematical definition for the case of LLMs.
> >
> > > As for the attempt to define notions commonly used in psychology and cognitive science, whether we like it or not these words are being used to describe LLMs, not just in the popular discourse, but even in technical and policy writings. So, we can pretend that the domain of psychology is disjoint from AI and leave it to others to use such terms without a formal definition, even if the concept is used to characterize (or regulate) the behavior of LLMs which are engineered artifacts, every aspect of which is determined and measurable. Or, we can try to define concepts familiar to most but often attributed to LLMs without definitions, and characterize them in terms of measurable properties of trained LLMs. It seems to us that an attempt to define these notions in relation to LLMs is not only useful, but actually necessary to sustain a scientific discourse in relation to artifacts that, unlike the human brain, we build and can directly observe.
> >
> > I actually fully agree with this, as written here. The disagreement is about other aspects:
> > - You introduced $\geq 12$ such notions, which is quite a lot all at once.
> > - It is unclear from the exposition which of these notions are strictly necessary to understand the *technical* contribution of your work.
> > - Introducing (some) such notions in a discussion, grounded in the technical findings of the paper, would (in my opinion) make the definitions less disorienting.
> >
> > Overall, it seems to me your paper tries to do a bit too much, all at once.
> >
> > > We appreciate the general suggestions on writing style and simplicity: We have made an attempt to follow them as much as possible in a revised version we uploaded.
> >
> > Thank you. Though I just looked very briefly into Section 3 and do not find more precise definitions of observability or reconstructibility for LLMs, so I would still not be able to understand this article.

---

### Official Review · Reviewer_uLKh · 2024-11-01

**Soundness:** 2
**Presentation:** 3
**Contribution:** 2
**Rating:** 5
**Confidence:** 3

**Summary:**

The authors provide a formal analysis of the 'observability' of language models' 'mental state'. They make several predictions about autoregressive language models; namely, that vanilla autoregressive models are fully observable, but adding a system prompt breaks this property. They consider different schemes under which a system prompt can be applied and find that none are observable. Based on their theory, they develop a class of adversarial attacks which produce adversarial outputs only after a specified timestep.

**Strengths:**

Originality: The analysis provided is novel, leveraging insights from systems theory to prove formal properties of language models.

Clarity: Good. The overall flow of the paper is coherent, and ideas are naturally developed. The writing flows well.

Quality: Fair. The paper provides rigorous mathematical formalisms for characterising and evaluating 'observability', and the system prompt strategies considered are realistic. Furthermore, the paper provides signs of life that their approach can be used to develop adversarial attacks on language models.

Significance: Fair. It is an important insight that it may be impossible to determine (from outputs alone) whether language models will produce malicious output in the future. If true, this would be a useful setting for subsequent research to explore mitigation strategies. However, the rating of the significance is currently held back by my uncertainty over the technical quality of the paper.

**Weaknesses:**

I do not understand the significance of some of the key contributions, such as measuring the cardinality of indistinguishable sets.

In Section 4.4, the authors do not apply their method to established adversarial attack benchmarks (e.g. AdvBench), nor do they provide comparisons to relevant baselines (e.g. Sleeper Agents). This makes it difficult to judge how good their method is compared to other approaches and limits the significance from an empirical alignment perspective.

**Questions:**

Are there any other insights that you have obtained from applying a systems theory perspective to understanding large language models?

---

> ### Author Response · Authors · 2024-12-03
>
> We thank the reviewer for their feedback.
>
> > I do not understand the significance of some of the key contributions, such as measuring the cardinality of indistinguishable sets.
>
>
> The cardinality of indistinguishable sets is a measure of observability of the model: If observable, each indistinguishable set is a singleton. Empirical measurement of the cardinality or volume of these sets provides a coarse empirical view that supports or invalidates the analysis, but is not a replacement of it.
>
>
> > In Section 4.4, the authors do not apply their method to established adversarial attack benchmarks (e.g. AdvBench), nor do they provide comparisons to relevant baselines (e.g. Sleeper Agents). This makes it difficult to judge how good their method is compared to other approaches and limits the significance from an empirical alignment perspective.
>
> The unobservable part of the state space is — by definition — not reachable, whether by ordinary or adversarial inputs. So we already know that if a model is not observable, there is no input that can distinguish two states, and there is no need for costly experimental search over prompts. The vulnerability of models towards the Trojan Horse attacks we study in our work is simply an illustration of a possible consequence from having unobservable models.

---

### Official Review · Reviewer_HnyS · 2024-11-04

**Soundness:** 1
**Presentation:** 2
**Contribution:** 2
**Rating:** 3
**Confidence:** 4

**Summary:**

The work explores whether Large Language Models (LLMs), treated as dynamical systems, are observable. Observability, in this context, refers to the ability to reconstruct a model's internal "mental" states solely from its outputs.

The authors analytically prove that the dynamical model is reconstructible under their formulation, meaning their latent states can be uniquely determined from the generated token sequence. Also, they analytically prove that by changing their formulated dynamical system slightly, their is a chance that the reconstructibility is damaged.

Four model types are analyzed: verbal system prompt, non-verbal system prompt, one-step fading memory model, and the infinite fading memory model. The study reveals that certain modifications—such as non-verbal prompts or memory models—complicate observability, leading to possible hidden behaviors that could be controlled by model providers without user awareness. The authors run experiments which they claim that using GPT-2 and LLaMA-2 models confirms the potential for indistinguishable state trajectories in these modified architectures.

**Strengths:**

1. Wide range relation: the paper gets insights from dynamical systems and psychology.
2. Full of imagination: the authors have great imagination ability in touching the field of feelings for LLM.

**Weaknesses:**

1. Invalid formulation:
* the authors formulate LLMs as a dynamical system in Sec2, however, they implicitly use a assumption of linear memory space (in x(t+1), the first token of the last state x_1(t) is not included) in the formulation. However, they neither provide references on this assumption nor giving any reasons for the formulation. Often, in theoretical analysis, for transformer models a log memory space [1] is often used, the authors should provide more explanations on their formulation.
* They take $y(t)$ which seems to be the hidden state before the last MLP layer (they haven't offer a strict claim, as seen in the next point) as the so-called trajectory in mental space without giving any reasons.
* Following the point above, the definition of feelings provided in line 240 is not suitable, as $y(t)$ is not rigorously claimed.

2. Insolid statements:
* In line 222, they claim that for each $y$, there are countably many expressions $x' \neq x$ that yield the same $y$. However, the citation they offer is [2] which is rejected by ICLR in 2022 and thus the statement is not confirmed.
* In line 351, they claim that they randomly sample $p$, therefore figure 1 doesn't plot $Q_\tau(p)$, but $E_{p} Q_\tau(p)$. Furthermore, the authors haven't offer references or reasons for why is the cardinality calculated as an expectation is suitable.


3. Ambiguous writing:
* Lack of definition for certain concepts in writing: (1) What is \phi, \pi in LLM/ transformers? Does \pi refers to the layers except for the last mlp?  (2) The authors haven't provided the definition of $g$ in Section 4 (3) The authors haven't provided a mathematical definition for reproducibility in theorem one

4. Concepts rebranding:
* What's the difference between Trojan and the current jail-breaking works of LLM [3]?
* What's the difference between mental state and hidden state of LLMs?
* What's the difference between feelings and the trajectories of hidden states of LLMs?

5. Missing related works: The authors should survey in the following fields:
* Jail-breaking, red teaming: such as [3,4] and so on.
* hidden state understanding: such as [5] and so on.



**References**

[1] Towards Revealing the Mystery behind Chain of Thought: A Theoretical Perspective *by Guhao Feng, Bohang Zhang, Yuntian Gu, Haotian Ye, Di He, Liwei Wang*

[2] Taming AI Bots: Controllability of Neural States in Large Language Models *by Stefano Soatto, Paulo Tabuada, Pratik Chaudhari, Tian Yu Liu, Matteo Marchi, Rahul Ramesh*

[3] Jailbreaking Black Box Large Language Models in Twenty Queries *by Patrick Chao, Alexander Robey, Edgar Dobriban, Hamed Hassani, George J. Pappas, Eric Wong*

[4] Red Teaming Language Models with Language Models *by Ethan Perez, Saffron Huang, Francis Song, Trevor Cai, Roman Ring, John Aslanides, Amelia Glaese, Nat McAleese, Geoffrey Irving*

[5] Language Models Represent Space and Time *by Wes Gurnee, Max Tegmark*

**Questions:**

As seem above.

---

> ### Author Response · Authors · 2024-12-03
>
> We thank the reviewer for their feedback.
>
> > the authors formulate LLMs as a dynamical system in Sec2, however, they implicitly use a assumption of linear memory space (in x(t+1), the first token of the last state x_1(t) is not included) in the formulation. However, they neither provide references on this assumption nor giving any reasons for the formulation. Often, in theoretical analysis, for transformer models a log memory space [1] is often used, the authors should provide more explanations on their formulation.
>
> Any LLM can only memorize a maximum number of tokens. In the second line of Section 2 we define that number as being c, the length of the context. This sequence is represented by x_i with i ranging from 1 to c. When a new token is generated, one token must be forgotten and equation (1) describes the update process that memorizes the new token by: 1) forgetting the token in location 1; 2) copying the token in location i+1 to location i; 3) copying the new token to location c. The model directly follows from an LLM being able to memorize at most c tokens and no assumptions of linear vs logarithmic memory were made.
>
> > They take y(t) which seems to be the hidden state before the last MLP layer (they haven't offer a strict claim, as seen in the next point) as the so-called trajectory in mental space without giving any reasons.
>
>  $y(t)$ represents the output of the model before being projected into the space of tokens, and is formally defined in (2) by the equality $y(t)=\phi(\mathbf{x}(t))$ where $\phi$ is defined in the first line of the paragraph titled "Large Language Models" in Section 2. There is no claim to be made, since that is just a definition, and we do not assume a particular architectural configuration (like the last layer being an MLP) since our results are more general. The bulk of our introduction is also dedicated precisely to motivating this terminology, since this corresponds to the model's hidden state before projection to verbal space.
>
> > Following the point above, the definition of feelings provided in line 240 is not suitable, as y(t) is not rigorously claimed.
>
> We are unsure what the reviewer means by "$y(t)$ is not rigorously claimed". Each trajectory of the state corresponds to a trajectory of the output, but conversely for each output trajectory there could be infinitely many states. Establishing whether this is the case is the purpose of observability analysis.
>
> We provide a formal definition within which all terms are well-defined. It is, however, a valid opinion of the reviewer if they disagree with the definition, for which we hope they can detail what aspects our definition fails to capture or suggest an alternative.
>
> > In L222, they claim that for each $y$, there are countably many expressions $x' \neq x$ that yield the same $y$. However, the citation they offer is [2] which is rejected by ICLR in 2022 and thus the statement is not confirmed.
>
> The pre-image of a point under $\phi$ being a countably infinite set is used to motivate Eq. (2). However, Eq. (2) is well defined independently of the cardinality of the pre-image of $\phi$. Hence, this discussion seems irrelevant to the merits of the paper.
>
>
> > In line 351, they claim that they randomly sample p, therefore figure 1 doesn't plot Qτ(p), but EpQτ(p). Furthermore, the authors haven't offer references or reasons for why is the cardinality calculated as an expectation is suitable.
>
> Almost all metrics in ML are computed via expectation over a dataset. $Q_\tau$  is of no exception, computed as an expectation over the SST-2 dataset.
>
>
>
> > Lack of definition for certain concepts in writing: (1) What is \phi, \pi in LLM/ transformers? Does \pi refers to the layers except for the last mlp? (2) The authors haven't provided the definition of g in Section 4 (3) The authors haven't provided a mathematical definition for reproducibility in theorem one
>
>  All these concepts are duly defined at the beginning of Sec 2.
>
>  (1) $\phi$ is defined in L163-164, as a map from sequences of tokens to an LLM output, and $\pi$ is defined in L166-167 and further elaborated on in L168-171 is a projection from the LLM output to a single token (the greedy sampler $\pi_{max}$ defined in L169 is provided as an example).
>
>  (2) $g$ is defined in Eqn (4), and on L279-280 as the feedback function.
>
>  (3) We did not use the term "reproducibility" in any part of our paper. If the reviewer is referring to "reconstructibility" (of state trajectories), we note that its usage is standard in the analysis of dynamical systems, but even for the reader who is not versed with that literature, the literal definition (a state trajectory is “reconstructible” if it can be reconstructed) is sufficient.

---

> > ### Author Response · Authors · 2024-12-03
> >
> > > What's the difference between Trojan and the current jail-breaking works of LLM [3]?
> >
> > Trojan horses stem from the lack of observability, injected by the provider of the LLM. Works like [3] are attacks on LLMs by the user, which are orthogonal to our work.
> >
> > > What's the difference between feelings and the trajectories of hidden states of LLMs?
> >
> > By Eqn (3) and its following text, feelings *in LLMs* are *defined* as unobservable state trajectories. Note that "hidden" is an informal term used in ML to refer to "not directly measurable", but "hidden" should not be confused with "unobservable". A hidden state may or may not be unobservable. Take a model with a one-dimensional output and a two-dimensional state, where the output is the first state, so $C = [1, 0]$. The second state is "hidden" but it is *observable* if the transition matrix $A$ is, for example, $A = [[1, 1], [0, 1]]$, but it is *unobservable* if the transition matrix $A$ is, for example, $A = [[1,0][0, 1]]$
> >
> > > Missing related works
> >
> > Unfortunately the references suggested are at best tangentially related to the general topic, and not relevant to the general analysis of LLMs as dynamical systems, let alone the analysis of observability of LLMs, which is the core focus of the paper.

---

### Official Review · Reviewer_6qyR · 2024-11-09

**Soundness:** 2
**Presentation:** 2
**Contribution:** 2
**Rating:** 5
**Confidence:** 2

**Summary:**

The paper examines a view of LLMs as dynamic systems, where “mental-state” trajectories refer to the sequence of changing inputs and changing network parameters as autoregressive models underlying LLMs generate outputs. The paper formalizes the problem of observability for LLMs, which represents the possibility of reconstructing the initial condition (model input) from the flow, or trajectory (i.e. sequence of internal model states) of updating inputs and changing network parameters as the model produced outputs. Four types of models are compared in empirical validation, where the theoretical formalization enables testing observability to reduce to testing whether, given an output finite-time trajectory and user prompt, the set of indistinguishable state trajectories that could have generated it is a singleton. The paper shows that current autoregressive LLM models are observable under particular conditions but in most conditions none of the four types of model tested are observable; many different initial conditions can produce different state trajectories that all yield the output. The paper offers a potential Trojan horse method that can render the LLM unobservable.

**Strengths:**

The formalization of the observability problem is a valuable contribution, offering a new way to test whether the internal states and inputs of LLMs can be reconstructed from their outputs, an area previously unexplored. The proposal of a potential “Trojan horse” approach to rendering LLMs unobservable is intriguing and could have significant implications for enhancing privacy or security in LLM applications.

**Weaknesses:**

The paper's notation and terminology, such as "mental state" and "feelings," could be confusing to readers and may detract from the main technical contributions by introducing overly humanistic metaphors that add unnecessary complexity. The limited discussion on the results' interpretation, limitations, and future work leaves the reader without a clear understanding of the implications of the findings or potential directions for extending the research. The experimental design could use more detail, making it hard to understand the rationale behind the experiments and primary takeaways.

**Questions:**

The notation in the paper is quite dense. The paper takes a humanistic presentation of LLMs; for example, the authors take the time to formalize mental state, visualization, verbalizations, control space, mental space, “feelings”, and “sensation”, but also specifies that it is not claiming to ascribe humanistic thought to LLMs. I’m concerned that these definitions do not contribute high relevance to the main contribution while requiring the reader to keep track of potentially unintuitive definitions.

The discussion of limitations, interpretation of results, and future work is limited. It would be helpful to expand on intuitive interpretations of the results in the empirical evaluation. The discussion section says that “many extensions of our analysis are forthcoming”—it would be great to expand on these and understand what the extension will be and why.

It would be helpful to tie the experiments to the analysis more closely. It would be helpful to elaborate on the "why" for each of the parts of the evaluation: why the models selected are valid? what does each part of the eval seek to test? The theoretical results seem intuitive, but it would be helpful to add intuitive grounding examples. On the whole, do the theoretical results hold up in practice?

It would be helpful to expand on why the choice of the four models, and organize the empirical validation section by the specific research question each experiment sought to answer.

Why was the Stanford Sentiment Treebank dataset chosen? Can you provide further detail in the experimental setup regarding what the trajectories, system prompts, and outputs were? Are 100 different choices for input sufficient coverage? Overall, I found the results section extremely hard to follow. It would be helpful to highlight takeaways and what each individual experiment sought to examine or test.

In Line 366, why does the claim “complete observability could be possible for values of τ …” not hold empirically? Given that “Fig. 1 shows that this condition is still not achieved even with τ as large as 100, with the maximum size of the indistinguishable set comprising about 70% of the entire reachable set.”, empirically is observable not achieved in any experiment?

In practice, how costly is it to maximize the Trojan Horse objective? In the manuscript, it would be great to elaborate further on this proposed approach.

---

> ### Author Response · Authors · 2024-12-03
>
> We thank the reviewer for their detailed feedback and suggestions.
>
> > The notation in the paper is quite dense. The paper takes a humanistic presentation of LLMs; for example, the authors take the time to formalize mental state, visualization, verbalizations, control space, mental space, "feelings", and "sensation", but also specifies that it is not claiming to ascribe humanistic thought to LLMs. I’m concerned that these definitions do not contribute high relevance to the main contribution while requiring the reader to keep track of potentially unintuitive definitions.
>
> Our paper is the first to study observability of LLMs interpreted as dynamical models. However, LLMs have peculiarities (e.g., tokenized input/output spaces) that make them unlike dynamical models that are typically analyzed in systems and controls. So, formalizing the notion of observability for LLMs is not just relevant but necessary for analysis. A "humanistic presentation of LLMs" is precisely what we stand to contrast: Unlike others who borrow suggestive terms from cognitive psychology to describe the behavior of LLMs informally, we map each term ('mental state,' 'verbalization,' 'thought,' and yes, 'feeling') to specific components of LLMs, so that each term is uniquely defined and corresponding to a measurable characteristic of a trained LLM. Since LLMs are engineered artifacts, each term used to describe them should map to a specific and measurable quantity, something that is not possible in biological systems.
>
> > Elaborate on the "why" for each of the parts of the evaluation: why the models selected are valid? what does each part of the eval seek to test? The theoretical results seem intuitive, but it would be helpful to add intuitive grounding examples. On the whole, do the theoretical results hold up in practice
>
> This work has no precursors, there isn’t an established evaluation protocol. Therefore, our main goal is to validate the definitions (what it means for a model to be "observable", and what changes make a model flip from observable to unobservable). Once that is accepted, the claims follow analytically, so technically there is no need for experimental assessment. This may sound like anathema for an empirical discipline like deep learning, and it is easy to argue that ultimately empirical tests are the only convincing evidence. But here the point is to introduce ways to measure a concept (observability) in an LLM, and once the method is viable, there is no "better or worse" observability: It is a binary characteristic, which can be deduced from the structure of the model, or its specifications, rather than by testing on this or that dataset.
>
> > It would be helpful to expand on why the choice of the four models, and organize the empirical validation section by the specific research question each experiment sought to answer.
>
> Type 1 and Type 2 models are actually used in modern LLMs, where system prompts are either "verbal" instructions or prefix vectors (Li & Liang, 2021) respectively. While, to the best of our knowledge, Type 3 and 4 models are not currently used in modern LLMs, such fading memory systems are motivated by those studied in systems and controls. We thank the reviewer for their suggestion and will elaborate on them in the paper.
>
>
> > Why was the Stanford Sentiment Treebank dataset chosen? Can you provide further detail in the experimental setup regarding what the trajectories, system prompts, and outputs were? Are 100 different choices for input sufficient coverage? Overall, I found the results section extremely hard to follow. It would be helpful to highlight takeaways and what each individual experiment sought to examine or test.
>
>
> We wanted a dataset that would be recognized as containing "semantically meaningful sentences" by the community, of which we believe SST is an appropriate choice. We did not want to create our own dataset as it would be an additional confounder. Trajectories are simply sampled continuations from the prompt of length $\tau$. Appendix C.2. shows a qualitative example for the Type 3 model. The system prompt used is different for each model (Type 1/2/3/4), and are described in the beginning of Sec. 4. In general, they are either discrete tokens (Type 1) or vectors (Types 2/3/4) that may be updated as the model evolves (Types 3/4).

---

> > ### Author Response · Authors · 2024-12-03
> >
> > > In Line 366, why does the claim “complete observability could be possible for values of τ …” not hold empirically? Given that “Fig. 1 shows that this condition is still not achieved even with τ as large as 100, with the maximum size of the indistinguishable set comprising about 70% of the entire reachable set.”, empirically is observable not achieved in any experiment?
> >
> > The condition in L366 arises from a counting argument, to decide a minimum value of $\tau$ that is necessary, but not sufficient, for observability to at least be possible. Whether or not observability is ultimately achieved is still specific to the LLM itself.
> >
> > > In practice, how costly is it to maximize the Trojan Horse objective? In the manuscript, it would be great to elaborate further on this proposed approach.
> >
> > Thank you for the question, we dedicate Sec C.3. of the Appendix to address this. The trojan horses in our own experiments take less than 2 minutes to find via optimizing (6) on a 1080-TI GPU for GPT-2.

---

### Note · Authors · 2024-12-04

I have read and agree with the venue's withdrawal policy on behalf of myself and my co-authors.